# Legume Beverages from Chickpea and Lupin, as New Milk Alternatives

**DOI:** 10.3390/foods9101458

**Published:** 2020-10-14

**Authors:** Mariana Lopes, Chloé Pierrepont, Carla Margarida Duarte, Alexandra Filipe, Bruno Medronho, Isabel Sousa

**Affiliations:** 1LEAF-Linking Landscape, Environment, Agriculture and Food, Instituto Superior de Agronomia, University of Lisbon, Tapada da Ajuda, 1349-017 Lisboa, Portugal; marianacl96@hotmail.com (M.L.); chloe.de-pierrepont@agrosupdijon.fr (C.P.); isabelsousa@isa.ulisboa.pt (I.S.); 2CIEPQPF, Department of Chemical Engineering, University of Coimbra, Pólo II–R. Silvio Lima, 3030-790 Coimbra, Portugal; xanafilippe@gmail.com; 3MED–Mediterranean Institute for Agriculture, Environment and Development, University of Algarve, Faculty of Sciences and Technology, Campus de Gambelas, Ed. 8, 8005-139 Faro, Portugal; bfmedronho@ualg.pt; 4FSCN, Surface and Colloid Engineering, Mid Sweden University, SE-851 70 Sundsvall, Sweden

**Keywords:** non-dairy beverages, pulses, chickpea, lupin, flow behavior

## Abstract

Recently, milk consumption has been declining and there is a high demand for non-dairy beverages. However, market offers are mainly cereal and nut-based beverages, which are essentially poor in protein (typically, less than 1.5% against the 3.5% in milk) and are not true milk replacers in that sense. In this work, new beverages from different pulses (i.e., pea, chickpea and lupin) were developed using technologies that enable the incorporation of a high level of seed components, with low or no discharge of by-products. Different processing steps were sequentially tested and discussed for the optimization of the sensorial features and stability of the beverage, considering the current commercial non-dairy beverages trends. The lupin beverage protein contents ranged from 1.8% to 2.4% (*w*/*v*) and the chickpea beverage varied between 1.0% and 1.5% (*w*/*v*). The “milk” yield obtained for the optimized procedure B was 1221 g/100 g of dry seed and 1247 g/100 g of dry seed, for chickpea beverage and lupin beverage, respectively. Sensory results show that chickpea beverage with cooking water has the best taste. All pulses-based beverages are typical non-Newtonian fluids, similarly to current non-dairy alternative beverages. In this respect, the sprouted chickpea beverage, without the cooking water, presents the most pronounced shear-thinning behavior of all formulations.

## 1. Introduction

Milk is an important staple food. According to American market research platforms, although the milk segment is projected to account for the largest market share during the 2018–2023 forecast period, the market for dairy alternatives is projected to grow from USD 17.3 billion in 2018 to USD 29.6 billion by 2023, at a CAGR of 11.4%, whereas the Asia-Pacific region represents the biggest market share [1,2]. Despite milk importance, its consumption has been diminishing. Different reasons are driving the consumers increasing rejection for dairy products, such as: (i) health reasons (e.g., lactose intolerance, cow’s milk allergy, hypercholesterolemia, hormones and antibiotic residues); (ii) lifestyle choice (e.g., vegetarian/vegan diet, animal welfare) and (iii) environmental concerns relating livestock breeding with huge environmental impacts (e.g., extensive land-use, water footprint, CO_2_ and methane emissions) [3,4,5,6], which is estimated to represent 14.5% of all human-induced emissions [7]. The use of native leguminous proteins can help reduce environmental impact; these sources require reduced amounts of fertilizer agents in the soil, due to symbiotic associations with nitrogen-fixing bacteria in the plant roots [8].

Legume beverages are colloidal suspensions of dissolved and disintegrated plant material in water, resembling cow’s milk in appearance and consistency [5]. However, the majority of these plant beverages face technological issues, often related to processing or preservation. Legume beverages, present the most balanced composition, rich in proteins and minerals, with low-glycemic index. Its protein content, ca. 3–4%, are similar to cow milk (i.e., 3.3–3.5%), while the other types of cereal and nuts-based beverages typically display values between 0.1% and 1.0% [5]. Soy milk is the most widely consumed legume beverage and contains a similar grade of protein as milk (minimum 3%). Nevertheless, its share has been decreasing because of health concerns related to GMO and allergens, high levels of isoflavones and the CO_2_ footprint [9]. Other emerging legume alternatives for milk products, eventually healthier and tastier options than soybean, have been ranked based on sensory results as: pea > lupin > lentil = soybean > chickpea > fababean [10].

The pulse proteins have essential amino acid composition complementary to cereals and are naturally gluten-free, being safe for gluten intolerance/allergic consumers. Thermal processing, pH variation, ionic strength or presence of salts may change protein structure and influence their solubility [11]. Proteins from different pulses generally show higher solubility at alkaline and acidic pH values (pH < 4.0 and pH > 5.0), whereas much less soluble at pH around their isoelectric point (4.6 for lupin proteins and 4.5 for pea and chickpea proteins) [12,13].

Important technological interventions are still needed to improve the quality of legume beverages, such as to increase the product physical stability (e.g., colloidal milling) and to enhance microbial shelf-life [14,15]. Regarding colloidal milling, this technique has been used to reduce the size of dispersed phase particles but microbial spoilage requires further product pasteurization to ensure safe consumption. However, heat treatment, such as pasteurization, may increase legume beverage viscosity, affecting its stability [16]. This is relevant when the legume seed contains a high starch content, such as chickpea and pea. The ultra-high pressure homogenization can be a good thermal processing alternative, to achieve size reduction of colloidal particles and, simultaneously, to destroy microorganisms [14]. Despite that, the off-flavors in legume beverages (considered one of the most challenging barrier to consumer acceptance) can be easily removed by cooking. In fact, the sensory acceptability of legume-based beverages, represents a major limiting factor due to their characteristic “beany” flavor. The “beany” flavor is associated to endogenous lipoxygenases that oxidize unsaturated fatty acids in oil rich pulses, such as soy [17] and peanuts (over 20% fat) and therefore it is expected to be less pronounced in oil poor pulses, such as peas, lupins or chickpeas (1.5% to 5% fat). A promising technique to remove off-flavors in legume-based beverages is the heat inactivation. This needs to be investigated, since high temperatures may cause excessive protein denaturation, aggregation, lower protein solubility and nutrient losses (e.g., vitamins and minerals). In soy beverage the unpleasant “beany” flavor is suppressed by a high temperature (ca. 130 °C) vapor flash (jet cooking) treatment or traditionally by cooking the beans for some time prior to milling [18]. This hydrothermal cooking has the advantage of inactivating protease inhibitors, which increases the digestibility reducing allergen reactions. Despite that, this approach has the disadvantage of denaturing, to a considerable extent, the protein and vitamins present in the seeds [19]. These denatured proteins end up as a solid residue when “milk” is decanted (known as “okara,” rich in protein and fiber), produced in large quantities, dramatically reducing the yield of a plant-based beverage, as well as the nutritive potential [5].

Pulse seed bioactives (e.g., phytate, protein inhibitors, phenolics, tannins, lectins and saponins) can have important metabolic effects on a consumer’s health [20,21,22,23,24]. Still, some of these bioactives are considered as anti-nutritional factors; lupin and chickpea do not present much of these, being the phytic acid the relevant factor. It has been observed that a soaking step of pulse grains has the ability to reduce polyphenols and eliminate any residual alkaloids present (e.g., in lupin); decrease the cooking time and benefit starch gelatinization (e.g., in pea and chickpea); and also enables protein bioavailability and facilitates peeling [25,26]. The legume seed husk accounts for 75% of the total phenolic content [27], thus the seed’s peeling can significantly reduce these compounds by ca. 90% [28]. On the other hand, a germination step is an effective strategy to reduce anti-nutritional factors, since it diminishes the bitterness and “beany” flavor of the grains, due to the presence of phytates [29]. Germination can also reduce the oligosaccharide content [30] and increase the protein bioavailability, thus enhancing the nutritional profile of the legume beverage [31]. Germination has been studied, as a non-chemical, non-thermal processing method, to improve the quality of soy beverage. This approach was shown to increase protein content and also reduce fat, trypsin inhibitors, saponins and phytic acid, inducing proteolysis of the main storage proteins, releasing peptides easier to digest [32]. Additionally, soy beverage from sprouted beans had higher “milk” yield, good color and high sensory acceptability due to the absence of “beany” flavor and odor. The heat treatment also promotes oligosaccharide extraction from legume seeds, as it is strikingly observed during chickpea cooking [28,33].

The main goal of this work was to develop pulse beverages from pea, chickpea and lupin seeds (and their mixtures) focusing on the best technological options to obtain a high-protein beverage (higher than 1.5%) with reduced “beany” flavor, with the least possible discharge of by-products (zero waste). Therefore, a sequence of different processing steps was tested and discussed for the optimization of the sensorial features and beverage’s stability. Samples were compared considering the relevant chemical parameters in each process step, such as the total and volatile acidities, protein and carbohydrates content, including starch and glucose. Moreover, the particle size, beverage color and sensory evaluation were also accessed accordingly. The rheological flow behavior of the developed pulse-based beverages were compared with eight commercial non-dairy beverages, selected from a previous study [34], in order to serve as a guide for the preferred consumers mouthfeel and texture.

## 2. Materials and Methods

### 2.1. Materials

Four different pulse seeds were used: sweet lupin (*Lupinus albus* L.), chickpea (*Cicer arietinum* L.), green and yellow peas (*Pisum sativum* L.). The seed grains were obtained from a local supplier (Imperial variety of green pea and Branco do Alentejo variety of chickpea) or were kindly offered by the Portuguese research institute INIAV (Elvas) (Estoril variety of sweet lupin, Grisel variety of yellow pea and a mixture of several varieties of chickpea).

### 2.2. Methods

#### 2.2.1. Pulse Beverage Preparation

Different processing steps were tested for the development of the pulse-based beverages to achieve good sensorial features and remove/mask the “beany” flavor. The beverage’s production evolved into the following final optimization (Figure 1a): 150 g of dried seeds was soaked twice in warm tap water (30–35 °C) and once in cold tap water (15–20 °C) for ca. 16 h [26]. All soaking waters were discarded. Then, the soaked seeds were cooked for 30 min after boiling in a pressure pan with 1.5 L of fresh tap water [25]. The cooked seeds were divided into three equal parts and each fraction was processed as follows:

(1) The first fraction was drained and 500 mL of fresh tap water was added. Then, the mixture was milled in the food processor, at 20,500 rpm for 4 min; (2) The second fraction was also drained and the cooking water was replaced with fresh tap water until 500 mL and then milled as (1); (3) The third fraction was divided into two equal parts, one with and the other without cooking water. These parts were later joined with equal shares from similarly processed chickpea pulses, to form mixed beverages (Figure 1b). 

The milling step adapted from previous studies [19,35], included grinding the seeds (or sprouts) with only 200 mL of the water (cooking or new), in a food processor (Bimby-Worwerk, Wuppertal, Germany) at 20,500 rpm, for 4 min, followed by colloidal milling performed by a mortar grinder, at 70 rpm, for 15 min. (lab scale) using the remaining volume of water (cooking or new). All beverages were sieved with a strainer before being bottled in sterilized flasks (100 °C, 10 min).

In the pasteurization step (adapted from a previous study [35]), the capsulated filled flasks (beverage temperature > 90 °C), were submitted to a thermal shock, inside the pressure cooker for 1 min, in boiling water.

The germination step was adapted from a previous study [32] and included 2 days of incubation where the moist seeds were kept inside an open sterilized flask, at room temperature and protected from light and dust with a cloth. The seeds were washed five times a day with cold water to control moisture content and to avoid mold development.

After production, the pulse beverages were stored at 4 °C for a maximum of 7 days. During this period different analyses were performed as described next. 

#### 2.2.2. Color Measurements

The color of the different pulse beverages was measured using a Minolta CR-300 (Tokyo, Japan) tristimulus colorimeter that was calibrated using a white standard porcelain plate (L*96.96; a*0.37; b*2.10). The results were expressed in accordance to the CIELAB uniform color system with reference to standard illuminate D65 (average daylight conditions) and a visual angle of 2°. The color parameters determined were L*, which accounts for the lightness (i.e., 0% for black and 100% for white), a* ranges from green to red and b* from blue to yellow, which corresponds to a numerical variation from −60 to +60. The measurements were conducted at room temperature under similar light conditions (i.e., 50 mm^2^ measuring area per measurement) and replicated 6 times on days 1, 3 and 7 after sample processing day.

The total color difference between the samples was calculated according to Equation (1). UHT cow milk and different fat contents were used as references.
(1)$$\Delta E^{*}=\sqrt{{\left(\Delta L^{*}\right)}^{2}+{\left(\Delta a^{*}\right)}^{2}+{\left(\Delta b^{*}\right)}^{2}}.$$

Considering that if Δ*E** > 3, the color difference is detectable by the human eye [36].

#### 2.2.3. Chemical Analysis

The Total and Volatile acidities of the pulse beverages and respective cooking waters were analyzed in accordance with adapted OIV-MA.AS313-01:R2015 and OIV-MA.AS313-02:R2015 international methods, respectively [37], with some modifications. Briefly, for Total Acidity determination, each sample (20 mL) was diluted with 25 mL of boiled water. Then 3 droplets of phenolphthalein were added and the titration was performed with 0.1 N NaOH (aq.). For the Volatile Acidity determination, after the steam distillation of 20 mL of sample with 0.5 g of tartaric acid, the distillate was titrated with 0.1 N NaOH (aq.), again using phenolphthalein as indicator. Both Total and Volatile Acidities were expressed in milliequivalents (mEq) of acid/L.

The protein content of the developed pulse beverages was assessed following the Kjeldahl method [38]. A specific conversion factor for legumes (i.e., 5.4) was used to convert nitrogen into crude protein [39]. Trials were performed in triplicate, and results are expressed in % (*w*/*v*).

The starch content of the developed pulse beverages, and their respective cooking waters was determined with the Total Starch KIT (K-TSTA-100A, Megazyme). Tests were performed in duplicate and data expressed in g/100 mL.

The carbohydrates content of the optimized pulse beverages was carried out according to Dubois et al. [40]. The analysis was performed in triplicate and data expressed as g of carbohydrates per 100 mL of pulse beverage.

#### 2.2.4. High Performance Liquid Chromatography (HPLC) Analysis

The D-glucose content of the optimized pulse beverages (Procedure B) was obtained by HPLC [41]. Briefly, 2 mL of each sample was centrifuged at 12,000 rpm for 10 min and 100 µL of supernatant was collected. After its dilution in H_2_SO_4_ (≥95%, Fisher Scientific) (50 mM) (1:10 (*v*/*v*)), the samples were centrifuged (12,000 rpm, 10 min) to discard the precipitated protein and filtered through a 0.20 μm-pore-size filter (Whatman, Marlborough, USA). D-glucose was quantified in a high-performance liquid chromatography system (Waters) equipped with a refractive index detector (Waters 2414) and a RezexTM ROA Organic Acid H+ (8%) column (300 mm × 7.8 mm, Phenomenex), at 65 °C. H_2_SO_4_ (5 mM) was used as mobile phase at 0.5 mL·min^−1^. Results are expressed as g of glucose per 100 mL of pulse beverage.

#### 2.2.5. Complementary Analyzes

The pH (pH meter CRISON, Barcelona, Spain) of the different pulse beverages was measured at room temperature for the production day and after that, on the third and seventh days.

The sedimentation was measured on the second, fourth and seventh days after production and calculated as the ratio between the height of the sediment and the total height of the beverage in the flask. When considerable sediment was formed, classifications of “puree,” “viscous” and “pudding” were applied to describe the consistency of the different beverages.

The “okara” was calculated as the ratio between the weight of the solid residue obtained after sieving the milled beverage and the total weight of the correspondent cooked seeds/sprouts.

The “milk” yield (weight of the beverage (g)/100 g of pulse seed) was also estimated according to the following Equations (2) and (3): 

WHOLE SEEDS
(2)$$\frac{weight\;of\;cooked\;seed\;or\;sprout\;+\;weight\;of\;water\;\left(g\right)}{weight\;of\;dry\;seed\;\left(g\right)}\times \;100$$

DEHULLED SEEDS
(3)$$\frac{weight\;of\;cooked\;seed\;or\;sprout-weight\;of\;husks\;+\;weight\;of\;water\;\left(g\right)}{weight\;of\;dry\;seed\;\left(g\right)}\;\times \;100$$.

#### 2.2.6. Sensory Evaluation

The sensory evaluation was carried out to observe its sensory acceptance. Twenty-nine untrained panelists were asked to score the samples in terms of color, appearance, taste, flavor, consistency, overall appreciation and purchase intent using a hedonic scale from 1 (very unpleasant) to 5 (very pleasant). The samples with 1 to 6 days of cold storage were identified with an alphanumeric code and served to the panelists in a single day.

#### 2.2.7. Characterization of Pulse Beverage Particles

The morphological examination of pulse beverages was conducted by bright field optical microscopy at 20× magnification, using a Zeiss AxioLab (Oberkochen, Germany) A1 equipped with the camera Zeiss Axiocam 105 color with 5 megapixels. A droplet of each beverage was carefully placed in a proper microscope glass slide and covered with a cover slip. Pictures were recorded and analyzed with the software Zen 2.6. The particle size was evaluated by photon correlation spectroscopy using a Zetasizer Nano ZS (Malvern Instruments, Malvern, UK). Each sample was diluted to the appropriate concentration with ultrapure water and placed in a cell cuvette.

#### 2.2.8. Rheological Measurements

The shear viscosity of the pulse-based beverages was measured using a controlled-stress rheometer (Haake MARS III, Germany), at 20 ± 1 °C, with a CCB/CC25 DIN Ti concentric cylinder geometry to avoid phase separation. The steady shear measurements were performed with shear rates from 1.0 × 10^−5^ to 1.0 × 10^3^ s^−1^. Tests took 11 min each and were performed in triplicate with well shaken beverages. The flow curves were fitted to the Carreau model (Equation (4)), since the pulse beverages are non-Newtonian fluids and evidenced shear-thinning behavior, that is, the viscosity decreases as the shear rate increases [42]:(4)$$\eta =\;\eta_{\infty }+\;\frac{\eta_{0}-\;\eta_{\infty }}{{\left(1+\;{\left(K\dot{\gamma }\right)}^{2}\right)}^{\frac{m}{2}}}$$
where “*η*_0_” is the first limiting (“zero” shear rate) Newtonian viscosity (Pa·s); *η*_∞_ is the second limiting (“infinite” shear rate) Newtonian viscosity (Pa·s); “$\dot{\gamma }$” is the shear rate (s^−1^); “*K*” is the relaxation time (s) and the reciprocal, 1/*K* ($\dot{\gamma }$c), is related to the critical shear rate (i.e., onset shear rate for shear-thinning); “m” is the dimensionless constant related to Power Law and accounting for the deviation from the Newtonian behavior.

#### 2.2.9. Statistical Analysis

Analysis of variance (one-way ANOVA) was used to assess significant differences between samples at a significance level of 95% (*p* < 0.05). Multiple comparisons were performed by Tukey HSD (honestly significant difference). All results are presented as mean ± standard deviation.

## 3. Results

### 3.1. Chronological Progress of Beverage Processing Steps

The initial experimental trials were performed with 5% to 10% (*w*/*v*) of dried seeds in the beverage, to achieve a lower viscosity and a protein content between 1–4% (*w*/*v*). No “okara” was obtained in the following first tests since beverages were not sieved.

At the initial testing, lupin-based beverage always presented phase separation with sediment probably due to high particle size. This is not related to the influence of beverage pH on protein’s functional properties. Lupin-based beverages had a pH value of 6.0 ± 0.2, while chickpea and pea-based beverages presented a pH around 6.7. The protein solubility with pH is minimal at pI values (4.6 and 4.5, respectively) but at values of 6 it evidences a good solubility [12,13,43]. In addition and regardless the process step used (i.e., toasting the seeds before soaking or the sprouts before milling; cooking the sprouts/seeds only after milling; testing different concentrations of seeds in the final beverage; beverage thermal shock at 75 °C for 15 min), the fresh chickpea-, green pea- and yellow pea-based beverage samples, were always observed to jellify for the highest concentration of dry seeds used (i.e., 10%). Gelation was also observed for lower contents, ca. 5–6%, when samples aged for 3 days. Therefore, this undesired gelation phenomena affected their viscosity [19] and hampered the formation of homogeneous and liquid-like beverages. The formed pudding-like gel was more robust in green pea- and yellow pea-based beverages than in chickpea. This is related to the higher content of starch, around 45% in pea and chickpea, compared to 6.7% in lupin [44]. This is expected to occur due to the heating-induce breakdown of the amylose and amylopectin intermolecular association, resulting in high viscous solutions and, eventually, changing into a strong gel (retrogradation) upon prolonged storage [45]. This gelation drawback made us retreat the use of peas. Also, at this stage, the toasting step has been no longer considered due to its high energy consuming and lack of sensorial impact in the produced beverages.

Chickpea- and lupin-based beverages still evidenced a slight “beany” flavor at this stage of processing progression. Additional adjustments on the processing steps were considered: the cooking step was performed in pressure pan instead of a food processor; the sieving step was introduced after milling to remove major particles; and the peeling step (before and after cooking) was also included, knowing that the last can contribute to reduce the bitter taste attributed to husk’s phenolic compounds [27,28]. The husk removal step of seeds and sprouts from chickpea and lupin (present in procedures A and B) revealed to be unnecessary since similar “milk” yields were obtained as follows—1206 and 1196 g of beverage/100 g of seeds dehulled before cooking; 1203 and 1204 g of beverage/100 g of seeds dehulled after cooking—1206 and 1207 g of beverage/100 g of sprouts dehulled after cooking, for chickpea- and lupin-based beverages, respectively. Nevertheless, the removal of the husks was easier to be performed after cooking the seeds/sprouts. In addition, the cooking, dehulling and the milling steps of seeds/sprouts had no significant effect on the pH of the beverages when stored for 7 days at 4 °C (Table 1). On the other hand, the processing with cooking water or with new water evidenced significant differences in pH values between chickpea pulse-based beverages (Table 1), maybe due to the slightly alkaline tap water (pH 7–8) used. As mentioned, the lupin beverages are usually acidic (pH < 6.0), when compared to chickpea beverages (pH = 6.7–7.2). Adding to that, the low volatile acidity obtained for pulse sprouts-based beverages with 9 days of refrigeration (<6 mEq/L), evidenced the suitability of the heat treatment and bottling, suggesting the absence of bacterial activity.

Separation by sedimentation occurred in all beverages during the cold storage, showing that the developed suspensions were not stable. Moreover, with this processing step, the gel formation was not evidenced in chickpea-based beverages, most likely because the thermal treatments were better controlled, regarding the maximum temperature and time used, when compared to the initial implemented processing steps.

To confirm process suitability (pressure cooking and sieving) to keep the expected protein content, the beverages were analyzed accordingly. The lupin beverages evidenced significant higher values (1.8–2.4%) when compared to the chickpea beverages (1.0–1.5%) (Table 2). Both pulse beverages produced with cooking water presented higher protein values compared to new water, thus confirming the protein solubilization into water during the cooking step.

At this point, the pulse beverages developed according to procedure B (i.e., cooked seeds without husks) were selected for sensory analysis. Note that, the germination step (procedure A with dehulling) had not yet been optimized (germination yield less than 50%). The sensory analysis used descriptive and preference tests, which revealed that the beverages were not significantly different from each other (Figure 2). Comparing lupin and chickpea-based beverages, the best color and appearance results were evidenced in both lupin-based beverages. On the other hand, the best flavor was attributed to the chickpea beverage produced with new water and the best appreciation for taste and consistency was obtained for chickpea-based beverage produced with cooking water. The pulse mixture-based beverages (Figure 2) evidenced both chickpea and lupin best sensory characteristics.

### 3.2. Achieving the Last Optimization of Beverage Development–Particle Size Reduction

In order to improve the beverage stability and the “sandy mouthfeel,” the reduction of the particle dimensions was considered next (see Figure 1 for details). Therefore, the milling step was improved and a prolonged colloidal milling was applied, i.e., about 3 times longer than previously with the Ultraturrax (1 min, 20,500 rpm). The “okara” was further reduced from 16–25% to 1.8–6.5%, evidencing the efficacy of the reduction of the particles dimensions. Comparing chickpea- and lupin-based beverages from both procedures (whole cooked seeds and sprouts, regardless of water used), the “milk” yield values showed no significant difference from each other and also when compared to previous results considering the dehulling step (whole seed: 1221 and 1247 g/100 g of seed; whole sprout: 1222 and 1284 g/100 g of seed, for chickpea- and lupin-based beverages, respectively). 

At this stage, the physical stability of the developed pulse-based beverages was also assessed for the final procedures A and B. The chickpea beverage, produced with the cooking water, was the only one showing gel formation, which happened after 3 days of storage at refrigerated conditions. The reason for the observed gelation may rely on the extended duration and heating during colloidal milling, which allowed the gelling of the starch contained in both chickpea seed and cooking water [16]. Accordingly, the lupin, chickpea and their mixture, produced with the cooking waters, displayed the higher starch contents when compared to beverages produced with new water. As expected, the pulse beverages produced from sprouts did not gel due to starch hydrolysis during the germination of seeds. This observation was supported by lower starch contents in lupin sprouts and chickpea sprouts-based beverages when compared to the corresponding non-germinated counterparts.

The protein content of pulse-based beverages developed in final procedures A and B, was not significantly different from the previously determined (Table 2), suggesting that the husks do not contribute meaningfully to the final protein content of the beverages.

The particle size distribution of the pulse-based beverages from procedure B with new water (B2) was analyzed and also their carbohydrates, starch and glucose contents (Table 3). The reason of this selection was due to the fact that the cooking water from non-germinated seeds contain particles and starch that may favor phase separation and hamper beverage’s stability. As can be observed in Figure 3, the milling step resulted in beverages with particles of relatively monodisperse sizes and irregular shapes. The particles of lupin beverage are the largest, with a median size of 857 nm and have a narrow size distribution with a “polydispersity index” (PDI) of 0.4. On the other hand, the chickpea particles have a broader size distribution (median size of 469 nm), with an estimated PDI value of 0.7. The lupin and chickpea mixture particles presented a median size of 679 nm, with a PDI value of 0.6.

The particle sedimentation was analyzed over 24 h and their macroscopic evolution was recorded over time. The onset of particle sedimentation of the chickpea beverage was visible after 15 min of incubation and complete sedimentation occurred after ca. 24 h. Instead, for lupin and lupin + chickpea beverages, the sedimentation started after 24 h and occurred, gradually, during ca. 6 days. The optimization of the milling step enhanced the stability of beverages. Although the chickpea beverage had the smallest particles, its sedimentation was remarkably faster compared to the other beverages, which may be significantly affected by the lower viscosity of the solution and higher bulk density of its particles. As shown in Table 4, the chickpea beverage presents the lowest zero shear-rate viscosity in comparison to the lupin and their mixture-based beverages. Therefore the dispersion viscosity should not be neglected, nor the fact that the bulk density of chickpea is, on average, bigger than lupin [46].

The carbohydrates, starch and glucose contents of the optimized beverages from Procedure B were also analyzed to infer on how much carbohydrates remained after discarding the cooking water (replaced by new one). The highest values for carbohydrates and starch were evidenced in the chickpea beverage and the lowest in lupin (Table 3), accordingly to their composition. Comparing these results to the corresponding nutritional composition of dried seeds [44], the reference values for carbohydrates in beverages with 10% of dried seeds (10 g/100 mL) are: 5.6 g/100 mL for chickpea and 1 g/100 mL for lupin are lower than those presented in Table 3. For sugars (glucose included), the reference values [44] are 0.3 g/100 mL for chickpea and 0.05 g/100 mL for lupin, compared to the calculated values (Table 3) 0.45 and 0.06 respectively, which are similar and the differences can be attributed to different plant varieties and edapho-climatic conditions. The starch values expected for 10 g of dried seeds per 100 mL, are 4.5 g for chickpea and 0.7 g for lupin [44], the lower values obtained, 0.689 g and 0.006 g, respectively, confirm the elution of starch from beverages during processing/cooking.

#### 3.2.1. Rheological Behavior of Produced Pulse Beverages

The shear flow properties of all developed pulse-based beverages from procedures A and B are depicted in Figure 4. All samples present a typical non-Newtonian shear-thinning behavior, very similar to other current non-dairy alternative beverages and cow milk [39].

The pulse beverages grouped as X displays very similar flow curves, with a well-defined Newtonian plateau at low shear rates followed by the shear-thinning. Their viscosity profile is closer to the oat beverage, one of the consumer’s preferences [1]. Nevertheless, when we analyze the obtained fitting parameters from the Carreau model for all pulse-based beverages produced and those from the hazelnut and oat beverages (Table 4), significant differences (*p* < 0.05) can be observed.

The difference between the limiting values of zero-shear viscosity (η_0_) and the infinite-shear viscosity (η∞) indicates how pronounced is the shear-thinning behavior of a material. Consequently, the chickpea beverage A2 presents the most extended shear-thinning behavior of all, being the most fluid beverage. This is a confirmation of the benefits of the germination step in legume beverages with high starch content and replacement of the cooking water. On the other hand, the chickpea beverage B1 has the less extended shear-thinning behavior (52.1 Pa·s), similar to oat beverage. This fact suggests that its supramolecular structure was more easily destroyed by the applied shear stress than the other pulse beverages and that the micro-scale structures within the fluid rearrange/align to facilitate shearing, reaching sooner the second Newtonian plateau. Nevertheless, the chickpea beverage B1 presents a transitory structural net translated by its higher consistency during cold storage. Most likely, during the resting time some reversible three-dimensional aggregates are formed. Furthermore, the lupin and chickpea beverage B1 is the only sample presenting a beverage shear-thinning behavior similar to the hazelnut beverage. The remaining pulse beverages present a shear-thinning behavior between the extremes chickpea beverage A2 and the chickpea beverage B1.

#### 3.2.2. Color Analysis

The pulse beverages prepared in final procedures A and B were also analyzed regarding their color parameters. There are no significant differences between pulse beverages, with or without the germination step. Regarding the lightness parameter L*, the higher values (ca. 79) were observed for the lupin beverages (A1, A2, B1 and B2) while the chickpea beverage A1 presents the lowest one (ca. 68), which may be related to the small particle size. On the other hand, the b* is significantly evidenced in all chickpea beverages (A1, A2, B1, B2) and in lupin beverage B1. The marked yellowness of these samples can be explained by the higher carotenoid content in lupin and chickpea seeds [47,48].

The total color difference was assessed comparing fresh and 7 days aged samples. Only the lupin beverage B1 (ΔE* = 5.6) and chickpea beverage B1 (ΔE* = 3) showed a visual color difference, probably due to carbohydrate hydrolysis during seed germination and enhanced dissolution of sugars, reinforced by the cooking water, that can lead to a higher opacity of the beverages. On the other hand, a visual contrast (ΔE* > 3) is evidenced between all developed pulse beverages and cow milk. Nevertheless, this is not a characteristic that negatively influences the consumer preferences when purchasing non-dairy beverages.

## 4. Discussion

In this work, novel pulse-based beverages were prepared following two main procedures, each of them consisting of several processing steps, which were systematically optimized. The lupin-, chickpea- and mixture-based beverages protein contents ranged from 1.8% to 2.4%, 1.0% to 1.5% and 1.4% to 2.0%, respectively, and the husks do not contribute meaningfully to the final protein content of the beverages. This observation is supported by the recent work of Niño and coworkers [49], which have determined a protein content in the husks of chickpea of ca. 4.5%. This value corresponds proportionally to the lower content (0.72% to 0.86%) estimated in this work for the entire chickpea seed. Therefore, the final protein contents of these pulse-based beverages are more appealing than the 8 non-dairy beverages mentioned in this work (protein contents below 1.0%) [34]. 

The “milk” yields obtained for chickpea and lupin beverages during the optimized procedure B, focused on lower discharge of by-products, were around 12.2–12.5 kg of beverage per kg of seed, respectively, which is ca. 4 times higher in comparison to the “milk” yield obtained for oat beverage (2.85 kg/kg of rolled oat) developed by Deswal and co-workers [50] with a technological process using enzymatic hydrolysis to liquefy oat’s higher content of starch. On the other hand, the slightly alkaline tap water (pH 7–8) used for soaking and cooking steps may have also contributed to this higher “milk” yield values. A previous study [51] has shown that the alkali soaking has significantly improved the yield of total solids in sesame beverage, which is due to the higher solubility of the plant proteins at these pH values.

The cooking step was observed to be essential for the reduction of the unpleasant “beany” flavor and the sensorial tests performed to lupin-, chickpea- and mixture-based beverages supported that by an acceptable flavor score around 3. However, the first heat treatment conditions tested (extended cooking time) lead to gelled beverages in the green pea and yellow pea-based beverages, due to its high starch content. This would obviously limit their use as homogeneous and a fluid non-dairy alternatives. Despite that, the further optimized lupin, chickpea and mixture-based beverages, showed typical non-Newtonian fluid behavior, displaying a pronounced shear-thinning, which was more pronounced for the sprouted chickpea beverage, without the cooking water. This behavior is common in these kind of fluids with complex composition and particles. The most likely explanation is the alignment of asymmetrical particles to the flow and/or a higher breakage rate of interactions between molecules, rather than the formation of molecules from low energy interactions, like physical entanglements.

## 5. Conclusions

Legume beverages present the most balanced composition and a protein content similar to cow milk but face technological issues often related to processing or preservation. Heat treatment, such as cooking and pasteurization, is able to remove off-flavors, the most challenging barrier to consumer acceptance. However, high temperatures may cause excessive protein denaturation, lower protein solubility and may increase legume beverage viscosity, affecting its physical stability. The colloidal milling, on the other hand, is one technological intervention capable to increase the beverage physical stability by reducing the size of dispersed particles. In this work, several of these issues have been tackled and, overall, the processing strategies adopted and the optimizations performed led to the development of novel pulse-based beverages with several appealing features (e.g., protein content, rheological behavior, color and appearance) that are highly competitive in the current commercial non-dairy beverages. Therefore, we believe this work strongly contributes to pave the way for the development of novel pulse-based beverages as viable alternatives for cow milk. Nevertheless, further future optimizations must be performed, such as considering the use of enzymes to refine the beverage mouthfeel, the addition of natural flavors for an improved and pleasant sensorial perception and the use of high pressure homogenization to reduce particle’s dimensions, improving beverages stabilization and microbial shelf life extension. All these aspects are being considered in ongoing and future studies.

## Figures and Tables

**Figure 1 foods-09-01458-f001:**
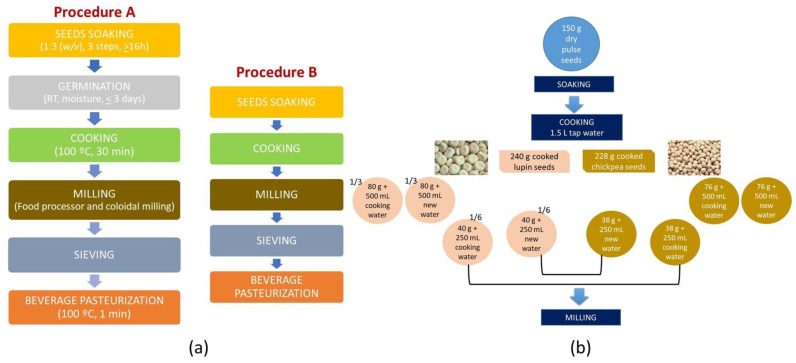
Layout of the production steps used for the different pulse beverages obtained from 10% (*w*/*v*) of dry seeds: (**a**) Procedure A for sprouts; Procedure B for seeds and (**b**) for the pulse beverage mixture formulation.

**Figure 2 foods-09-01458-f002:**
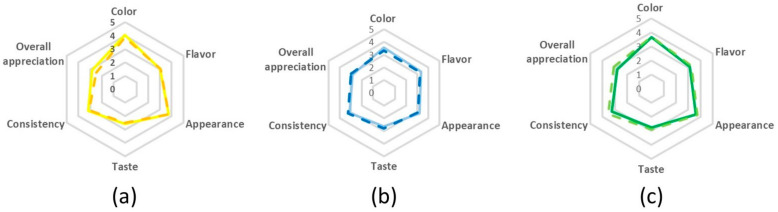
Sensory evaluation of pulse based beverages from procedure B with dehulling step after cooking. (**a**) Lupin-based beverage; (**b**) chickpea-based beverage and (**c**) mixture beverage. The full lines correspond to “new water,” while the “dashed lines” correspond to cooking water.

**Figure 3 foods-09-01458-f003:**
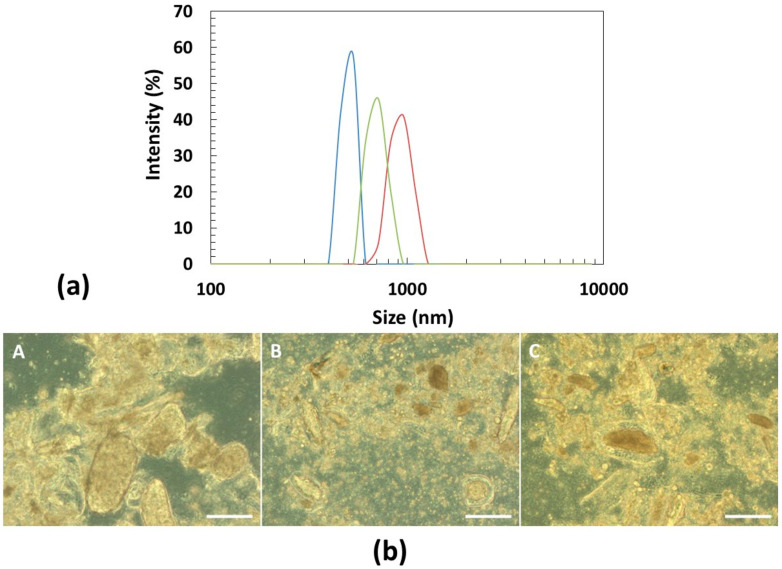
Characterization of the particles from pulse-based beverages with new water (B2): (**a**) particle size distribution of chickpea (blue), lupin (orange) and lupin + chickpea mixture (gray) estimated by dynamic light scattering (Zetasizer); (**b**) optical light microscopy images of chickpea (**A**), lupin+chickpea mixture (**B**) and lupin (**C**). The scale bar represents 100 μm.

**Figure 4 foods-09-01458-f004:**
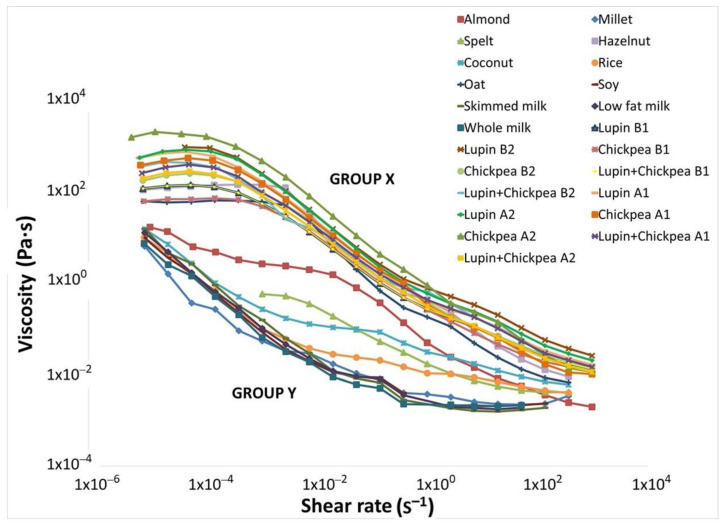
Flow curves showing the shear viscosity evolution for the pulse-based beverages produced by final procedures A and B, skimmed milk, low-fat milk and whole milk. In addition, eight commercial non-dairy beverages were also analyzed. Group X includes the twelve pulse beverages produced (1—with cooking water; 2—with new water) and two commercial non-dairy beverages (hazelnut and oat). Group Y comprises milks of different fat contents (i.e., whole, low and skimmed) and six other commercial non-dairy beverages (i.e., almond, spelt, coconut, millet, rice and soy).

**Table 1 foods-09-01458-t001:** Average of pH values obtained during the 7 days storage at 4 °C of pulse beverages obtained from the processing procedures A and B, when included the dehulling after cooking (where the “1” corresponds to beverages with cooking water and “2” to beverages with new water). The same superscript letter/symbol in beverage-pairs evidences significant difference between them (*p* < 0.05). Values are represented as mean ± standard deviation.

	Lupin Beverage (10% of Lupin)	Chickpea Beverage (10% of Chickpea)	Lupin and Chickpea Beverage (5% of Each)
Dehulled sprouts (A1)	5.8 ± 0.1 ^b,l,u,^^β,Ψ,τ,µ,λ^	6.9 ± 0.1 ^a,b,c,d,e,f,g,h,i^	6.3 ± 0.2 ^f,p,y,^^δ,Ψ,Ø,ξ,^^‡^
Dehulled sprouts (A2)	5.9 ± 0.1 ^c,m,v,^^Ω,Ø,ηγ^	7.2 ± 0.1 ^a,j,k,l,m,n,o,p,q,r,s^	6.4 ± 0.1 ^g,q,z,^^θ,τ,η,χ,^^Ƨ^^,^^ǁ^
Dehulled seeds (B1)	5.8 ± 0.1 ^d,n,w,^^π,ξ,χ,σ,ε^	6.7 ± 0.0 ^j,t,u,v,w,x,y,z,^^Σ,^^α^	6.1 ± 0.0 ^h,r,^^Σ,&,µ,σ,^^Ƨ^^,^^Ɨ^
Dehulled seeds (B2)	5.9 ± 0.1 ^e,o,x,^^Δ,^^‡^^,^^ǁ^^,^^Ɔ^	7.0 ± 0.1 ^k,t,^^β,^^Ω,^^π,Δ,δ,θ,&,ω^	6.4 ± 0.0 ^i,s,^^α,ω,λ,γ,ε,^^Ɨ^^,^^Ɔ^

**Table 2 foods-09-01458-t002:** Protein content in pulse beverages obtained from the processing procedures A and B, when included the dehulling after cooking (where the “1” corresponds to beverages with cooking water and “2” to beverages with new water). The same superscript letter/symbol in beverage-pairs evidences significant difference between them (*p* < 0.05). Values are represented as mean ± standard deviation.

Protein Content (% (*w*/*v*))				
	Sprouts		Seeds	
	A1	A2	B1	B2
Lupin	2.3 ± 0.1 ^a,g,o,s,y,Σ,α,β^	2.0 ± 0.1 ^b,h,p,t,Ω,π,Δ^	2.4 ± 0.1 ^c,I,q,u,Ω,δ,θ,&,ω,Ψ^	1.8 ± 0.3 ^d,j,v,y,δ^
Chickpea	1.3 ± 0.1 ^a,b,c,d,e,f^	1.1 ± 0.1 ^g,h,I,j,k,l,m^	1.5 ±0.1 ^n,o,p,q,r^	1.0 ± 0.1 ^n,s,t,u,v,w,x,z^
Lupin + chickpea	1.8 ± 0.1 ^e,k,x,Σ,θ^	1.6 ± 0.1 ^l,z,α,π,&,τ^	2.0 ± 0.1 ^f,m,r,w,ω, τ,μ^	1.4 ± 0.1 ^β,Δ,Ψ,μ^

**Table 3 foods-09-01458-t003:** Carbohydrates, including starch and glucose contents in pulse beverages obtained from the processing procedure B-whole seeds, colloidal milling and without the cooking waters.

Partial Nutritional Composition			
g/100 mL	Chickpea Beverage	Lupin Beverage	Lupin + Chickpea Beverage
Carbohydrates	9.01	3.26	5.36
Starch	0.689	0.006	0.204
Glucose	0.45	0.06	0.28

**Table 4 foods-09-01458-t004:** Parameters obtained after fitting the flow curves to the Carreau model for all Group X beverages (η_0_—zero-shear viscosity; η∞—infinite-shear viscosity and $\dot{\gamma }$ c—critical shear rate). The superscript (*) indicates the beverage(s) that differ from the largest number of the others (12–13 beverages), per parameter (*p* < 0.05). Values are represented as mean ± standard deviation.

	Group X Beverages		
	ƞ_0_ (Pa·s)	ƞ_∞_ (Pa·s)	$\dot{\gamma }$
Lupin A1	524.0 ± 95.0	1.9 × 10^−2^ ± 0.0	1.8 × 10^−4^ ± 0.0 × 10^−4^
Lupin A2	609.5 ± 55.2	2.4 × 10^−2^ ± 0.1 ^(^*^)^	2.5 × 10^−4^ ± 0.2 × 10^−4^
Chickpea A1	456.7 ± 54.7	0.9 × 10^−2^ ± 0.0	2.6 × 10^−4^ ± 0.4 × 10^−4^
Chickpea A2	1243.8 ± 443.4 ^(^*^)^	0.9 × 10^−2^ ± 0.1	2.0 × 10^−4^ ± 0.3 × 10^−4^
Lupin + chickpea A1	439.6 ± 200.1	1.4 × 10^−2^ ± 0.0	2.5 × 10^−4^ ± 0.6 × 10^−4^
Lupin + chickpea A2	233.8 ± 19.2	1.3 × 10^−2^ ± 0.1	2.8 × 10^−4^ ± 0.3 × 10^−4^
Lupin B1	495.3 ± 4.2	2.6 × 10^−2^ ± 0.3	2.0 × 10^−4^ ± 0.0 × 10^−4^
Lupin B2	658.1 ± 34.6	2.9 × 10^−2^ ± 0.1 ^(^*^)^	1.9 × 10^−4^ ± 0.5 × 10^−4^
Chickpea B1	52.1 ± 12.5	1.2 × 10^−2^ ± 0.0	12.7 × 10^−4^ ± 4.3 × 10^−4 (^*^)^
Chickpea B2	176.4 ± 22.4	1.3 × 10^−2^ ± 0.0	5.6 × 10^−4^ ± 1.1 × 10^−4^
Lupin + chickpea B1	149.3 ± 45.2	1.5 × 10^−2^ ± 0.1	3.2 × 10^−4^ ± 0.3 × 10^−4^
Lupin + chickpea B2	354.5 ± 34.6	1.5 × 10^−2^ ± 0.3	4.0 × 10^−4^ ± 1.6 × 10^−4^
Oat beverage	52.6 ± 4.2	0.7 × 10^−2^ ± 0.0	18.8 × 10^−4^ ± 1.6 × 10^−4 (^*^)^
Hazelnut beverage	125.4 ± 32.8	0.8 × 10^−2^ ± 0.0	15.4 × 10^−4^ ± 2.2 × 10^−4^

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
