# Peer review of "Legume Beverages from Chickpea and Lupin, as New Milk Alternatives"

_foods, 2020, doi:10.3390/foods9101458_

Round 1
Reviewer 1 Report
This manuscript investigates various procedures to prepare pulse-based beverages from chickpea and lupin. Those beverages were characterized on their composition, pH, viscosity, morphology and a sensory evaluation was conducted. The manuscript covers several interesting aspects that could contribute to new applications of these pulses in plant-based beverages. However, a clear focus seems to be lacking as well as an approach to get insight on the fundamental differences between the pulses and how this reflect their behaviour. For instance, the type of proteins (e.g. α-, β-, γ- and δ-conglutins in lupin), their isoelectric points or their functionality found in other studies, have merely been discussed or referred to. In addition, there is nothing mentioned on the bio-availability of the different proteins sources relative to milk derived protein. The major revisions are required in which the above-mentioned general comments, and the specific comments below, are taken into account.
Comments on specific issues:
- There are some spelling errors, for instance L72 “The seed husks accounts”, L191 “on triplicate”, L351 “that allowing”. Please check your manuscript on this.
-L82 the main aim of this work was to develop pulse beverages with the highest level of beneficial seed components. However, only the protein content was analysed. It this is the aim of your work it may be relevant to also quantify other nutrients such as fats and sugars.
-L84 An additional aim was to avoid extensive protein denaturation, but the protein denaturation has not been determined. It would be suggested to either measure whether the proteins were denatured with for instance DSC or CD, or to reconsider whether your initial aim matches your results.
-L153 a nitrogen conversion factor of 6.25 is rather high for plant-based proteins. It would be advised to use a value that averages the N-conversion factors of chickpea and lupin proteins or calculate the protein contents from the individual N-conversion factors.
-L187 how did the authors deal with sedimentation during these measurements. Did sedimentation occur within the time scale of the measurement and could it have affected rheological results?
-L191 a shear rate of 1.0x10-5 is extremely low. You may need to check the value of torque in the first part of your shear ramp is indeed above its minimal value.
-L217 different processing steps (e.g. toasting seeds before soaking, cooking sprouts / seeds after milling) did not affect the pH of the pulse-based beverages. Do the subsequent pH differences affect the viscosity or other properties? The author thoughts / a reference on how pH differences affect the viscosity of lupin and chickpea beverages would be in place.
-L240 what is the reason for keeping the seed particles in your dispersions? It seems that sedimentation could easily be avoided by removing larger particles (e.g. centrifugation, filtration) after solubilization of the protein. From an application point of view, it could also be more relevant if the dispersions were characterized without larger particles present.
-L279 insert ‘one’ between ‘only showing’
-L316 Can you relate the faster sedimentation of the chickpea particles to the viscosity of the dispersion? Also, you could consider determining the bulk density of the particles to substantiate your statement.
-L323 This graph (Fig. 4) is hard to read for comparison of individual samples.
Author Response
Carla Margarida Duarte
Instituto Superior de Agronomia
Tapada da Ajuda
1349-017, Lisboa, Portugal
Tel. (+351) 21 365 31 00
Fax: (+351) 21 365 31 95
E-mail: carladuarte@isa.ulisboa.pt
Dear Sirs,
We are pleased to re-submit herewith our revised manuscript entitled “Legume beverages from chickpea and lupin, as new milk alternatives” with Foods reference 933617.
We would like to thank the Referees for their constructive comments and criticism on the manuscript. All their suggestions were considered and included in the attached revised manuscript (main changes highlighted in different colour). The answers to the individual comments/questions are provided as attachment and we believe have significantly increased the quality of the manuscript.
Looking forward to hearing from you in due course.
Yours sincerely,
Carla Margarida Duarte (on behalf of all coauthors)
REVIEWER 1
Comments and Suggestions for Authors
This manuscript investigates various procedures to prepare pulse-based beverages from chickpea and lupin. Those beverages were characterized on their composition, pH, viscosity, morphology and a sensory evaluation was conducted. The manuscript covers several interesting aspects that could contribute to new applications of these pulses in plant-based beverages. However, a clear focus seems to be lacking as well as an approach to get insight on the fundamental differences between the pulses and how this reflect their behaviour. For instance, the type of proteins (e.g. α-, β-, γ- and δ-conglutins in lupin), their isoelectric points or their functionality found in other studies, have merely been discussed or referred to. In addition, there is nothing mentioned on the bio-availability of the different proteins sources relative to milk derived protein. The major revisions are required in which the above-mentioned general comments, and the specific comments below, are taken into account.
The Introduction section was extensively revised taking into account the reviewer valuable suggestions. The other general comments were also addressed and are discussed in the following answers.
All reference´s numbers in the manuscript and the list counting were corrected accordingly taking into account the new references introduced.
Comments on specific issues:
1- There are some spelling errors, for instance L72 “The seed husks accounts”, L191 “on triplicate”, L351 “that allowing”. Please check your manuscript on this.
All manuscript was revised for spelling errors as requested.
2- L82 the main aim of this work was to develop pulse beverages with the highest level of beneficial seed components. However, only the protein content was analyzed. It this is the aim of your work it may be relevant to also quantify other nutrients such as fats and sugars.
The sentence was re-phrased (new L152-161): “The main goal of this work was to develop pulse beverages from pea, chickpea and lupin seeds (and their mixtures) focusing on the best technological options to obtain a high-protein beverage (higher than 1.5%) with reduced “beany” flavor, with the less possible discharge of by-products. So, a sequence of different processing steps were tested and discussed for the optimization of the sensorial features and beverage´s stability. Samples were compared considering the relevant chemical parameters in each process step, such as the total and volatile acidities, protein and carbohydrates content, including starch and glucose. Moreover, the particle size, beverage color and sensory evaluation were also accessed accordingly. The rheological flow behavior of the developed pulse-based beverages was compared with eight commercial non-dairy beverages, selected from a previous study [39], in order to serve as a guide for the preferred consumers mouthfeel and texture.”
Following the reviewer suggestion, a new chemical analysis was included (new L301-303): “The carbohydrates content of the optimized pulse beverages was carried out according to Dubois et al. [45]. The analysis were performed in triplicate and data expressed as g of carbohydrates per 100 mL of pulse beverage.”
A new characterization method was included in the materials and methods section (new L304-312): “2.2.4 High performance liquid chromatography (HPLC) analysis. The D-glucose content of the optimized pulse beverages (Procedure B) was obtained by HPLC [46]. Briefly, 2 mL of each sample was centrifuged at 12.000 rpm for 10 min, and 100 µL of supernatant was collected. After its dilution in H2SO4 (98 %, Merck) (50 mM) (1:10 (v/v)), the samples were centrifuged (12.000 rpm, 10 min) to discard the precipitated protein, and filtered through a 0.20 μm-pore-size filter (Whatman). D-glucose was quantified in a high-performance liquid chromatography system (Waters) equipped with a refractive index detector (Waters 2414) and a RezexTM ROA Organic Acid H+ (8%) column (300 mm × 7.8 mm, Phenomenex), at 65 °C. H2SO4 (5 mM) was used as mobile phase at 0.5 mL.min-1. Results are expressed as g of glucose per 100 mL of pulse beverage.”
New references were added:
[45] Dubois, M.; Gilles, K.; Hamilton, J.K.; Rebers, P.A.; Smith, F. A colorimetric method for the determination of sugars and related substances. Nature 1951, 168, 167. https://doi.org/10.1038/168167a0
[46] Santos, M.V.; Faria, N.T.; Fonseca, C.; Ferreira, F.C. Production of mannosylerythritol lipids from lignocellulose hydrolysates: tolerance thresholds of Moesziomyces antarcticus to inhibitors. J Chem Technol Biotechnol 2019, 94, 1064–107. https://doi.org/10.1002/jctb.5853
New L762-765: “Table 3. Carbohydrates, including starch and glucose contents in pulse beverages obtained from the processing procedure B - whole seeds, colloidal milling and without the cooking waters.”
A new paragraph was added (new L766-778): “The carbohydrates, starch and glucose contents of the optimized beverages from Procedure B were also analyzed to infer on how much carbohydrates remained after discarding the cooking waters (replaced by new one). The highest values for carbohydrates and starch were evidenced in the chickpea beverage and the lowest in lupin´s (Table 3), accordingly to their composition. Comparing these results to the correspondent nutritional composition of dryed seeds [49], the reference values for carbohydrates in beverages with 10% of dryed seeds (10 g/100 mL) are: 5.6 g/100 mL for chickpea and 1 g/100 mL for lupin are lower than those presented in Table 3. For sugars (glucose included) the reference values [49] are 0.3 g/100 mL for chickpea and 0.05 g/100 mL for lupin, comparing to the calculated values (Table 3) 0.45 and 0.06 respectively, which are similar and differences can be attributed to different plant varieties and edapho-climatic conditions. The starch values expected for 10 grams of dried seeds per 100 mL, are 4.5 g for chickpea and 0.7 g for lupin [49], the lower values obtained, 0.689 g and 0.006 g, respectively, confirm the elution of starch from beverages during processing/ cooking.”
It is important to note that the analysis of fat content was not considered because according to INSA nutritional composition of the studied pulse seeds [49], the total lipid content of the chickpea seed is ca. 5% (w/w) and the lupin seed is ca. 2.4% (w/w). Since our pulse-based beverages contain only 10% (w/v) of seeds, this would represent an expected very low fat content in chickpea-based beverages (estimated to be 0.5%) and in lupin-based beverages (estimated to be 0.24%).
3-L84 An additional aim was to avoid extensive protein denaturation, but the protein denaturation has not been determined. It would be suggested to either measure whether the proteins were denatured with for instance DSC or CD, or to reconsider whether your initial aim matches your results.
We agree with the reviewers and therefore, the abstract was also reviewed (new L18-32): “Recently, milk consumption has been declining and there is a high demand for non-dairy beverages. However, market offers are mainly cereal and nut-based beverages, which are essentially poor in protein (typically, less than 1.5% against the 3.5% in milk) and are not true milk replacers in that sense.
In this work, new beverages from different pulses (pea, chickpea and lupin) were developed using technologies enabling the incorporation of a high level of seed components, with low or no discharge of by-products. Different processing steps were sequentially tested and discussed for the optimization of the sensorial features and stability of the beverage, taking into account the current commercial non-dairy beverages. The lupin beverage protein contents ranged from 1.8 to 2.4% and the chickpea beverage varied between 1.0 and 1.5%. The “milk” yield of the optimized procedure B was 1121 g/100 g of dry seed and 1247 g/100 g of dry seed, for chickpea beverage and lupin beverage, respectively. Sensory results shows that chickpea beverage with cooking water had the better taste. All pulse-based beverages are typical non-Newtonian fluids, similarly to current non-dairy alternative beverages. In this respect, the sprouted chickpea beverage, without the cooking water, presents the most pronounced shear-thinning behavior of all formulations.”
4-L153 a nitrogen conversion factor of 6.25 is rather high for plant-based proteins. It would be advised to use a value that averages the N-conversion factors of chickpea and lupin proteins or calculate the protein contents from the individual N-conversion factors.
The reviewer is right and the sentence was re-phrased (new L295-297): “The protein content of the developed pulse beverages was assessed following the Kjeldahl method [43]. A specific conversion factor for legumes (5.4) was used to convert nitrogen into crude protein [44]. Trials were performed in triplicate.”
New reference was added:
[44] Mariotti, F.; Tomé, D.; Mirand, P.P. Converting Nitrogen into Protein - Beyond 6.25 and Jones' Factors'. Crit Rev Food Sci 2008, 48, 2, 177 – 184. http://dx.doi.org/10.1080/10408390701279749
Table 2 (new L519-523) shows the corrected protein contents with the conversion factor of 5.4.
5-L187 how did the authors deal with sedimentation during these measurements. Did sedimentation occur within the time scale of the measurement and could it have affected rheological results?
The sentence was re-phrased (new L355-359): “The shear viscosity of the pulse-based beverages was measured using a controlled-stress rheometer (Haake MARS III, Germany), at 20 + 1 ºC, with a CCB/CC25 DIN Ti concentric cylinder geometry to avoid phase separation. The steady shear measurements were performed with shear rates from 1.0x10-5 to 1.0x103 s-1. Tests took 11 min each and were performed in triplicate with well shaken beverages.”
6-L191 a shear rate of 1.0x10-5 is extremely low. You may need to check the value of torque in the first part of your shear ramp is indeed above its minimal value.
This is an important point raised by the reviewer. The CCB/CC25 DIN Ti concentric cylinder geometry used in our trials was automatically recognized by the Haake MARS III software which calibration settled a minimum torque value and performed the necessary torque adjustments to achieve credible results. If the torque was below the equipment capacity, the software would display an error message which was not the case.
7-L217 different processing steps (e.g. toasting seeds before soaking, cooking sprouts / seeds after milling) did not affect the pH of the pulse-based beverages. Do the subsequent pH differences affect the viscosity or other properties? The author thoughts / a reference on how pH differences affect the viscosity of lupin and chickpea beverages would be in place.
The sentence was re-phrased (new L399-403): “At the initial testing, lupin-based beverage always presented phase separation with sediment (data not shown) must probably due to high particle size. This is not related to the influence of beverage pH on protein’s functional properties. The lupin-based beverages had a pH value of 6.0 + 0.2, while chickpea and pea-based beverages presented a pH around 6.7. The protein solubility with pH shows a minimum between 4 and 5, but at values of 6 it evidences a good solubility [12,48].”
The sentence was re-phrased (new L409-415): “Therefore, this undesired gelation phenomena affected their viscosity [21] and hampered the formation of homogeneous and liquid-like beverages. The formed pudding-like gel was more robust in green pea- and yellow pea-based beverages than in chickpea. This is related to the higher content of starch, around 45% in pea and chickpea, compared to 6.7% in lupin [49]. This is expected to occur due to the heating-induce breakdown of the amylose and amylopectin intermolecular association, resulting in high viscous solutions and, eventually, changing into a strong gel (retrogradation) upon prolonged storage [50]. This gelation drawback made us retreat the use of peas.”
New references were added:
[12] Piornos, J.A; Burgos-Díaz, C.; Ogura, T.; Morales, E.; Rubilar, M.; Maureira-Butler, I.; Salvo-Garrido, H. Functional and physicochemical properties of a protein isolate from AluProt-CGNA: A novel protein-rich lupin variety (Lupinus luteus). Food Res Int 2015, 76, 719-724. https://doi.org/10.1016/j.foodres.2015.07.013
[48] Sousa, I.M.N.; Morgan, P.J.; Mitchell, J.R.; Harding, S.E.; Hill, S.E. Hydrodynamic characterization of lupin proteins: solubility, intrinsic viscosity, and molar mass. J Agr Food Chem 1996, 44, 3018-3021. http://dx.doi.org/10.1021/jf950516f
[50] Tako, M.; Tamaki, Y.; Teruya, T.; Takeda, Y.. The Principles of Starch Gelatinization and Retrogradation. Food Nutr Sci 2014, 5, 280-291. http://www.scirp.org/journal/fns) http://dx.doi.org/10.4236/fns.2014.53035
8-L240 what is the reason for keeping the seed particles in your dispersions? It seems that sedimentation could easily be avoided by removing larger particles (e.g. centrifugation, filtration) after solubilization of the protein. From an application point of view, it could also be more relevant if the dispersions were characterized without larger particles present.
The reason for keeping the seed particles is to suppress losses and by-products or residues during processing.
New L266-268: Figure 1 was completed with the Sieving step between colloidal milling and pasteurization.
New sentences were added:
New L255-256: “All beverages were sieved with a strainer before being bottled in sterilized flasks (100 ºC, 10 min).”
New L396-398: “The initial experimental trials where performed with 5% to 10% of dried seeds. The goal was to achieve a final beverage with lower viscosity and a protein content between 1-4%. No “okara” was obtained in the following first tests since beverages were not sieved.”
New L418-423: “Chickpea- and lupin-based beverages still evidenced a slight “beany” flavor at this stage of processing progression. Additional adjustments on the processing steps were considered: the cooking step was performed in pressure pan instead of food processor; the sieving step was introduced after milling to remove major particles; and the peeling step (before and after cooking) was also introduced, knowing that the last can contribute to reduce the bitter taste attributed to husk’s phenolic compounds [28,33].”
The sentence was re-phrased (new L671-672): “The “okara" was further reduced from 16-25% to 1.8-6.5%, evidencing the efficacy of particles’ dimension reduction.”
9-L279 insert ‘one’ between ‘only showing’
Revised as requested (new L679).
10-L316 Can you relate the faster sedimentation of the chickpea particles to the viscosity of the dispersion? Also, you could consider determining the bulk density of the particles to substantiate your statement.
We appreciate the reviewer question since the viscosity on the dispersion is expected to play an important role on the rate of sedimentation of the particles. From Stokes, we know the rate of sedimentation depends on the particle size, gravity force, fluid viscosity and density difference between particle and fluid. Taking this into account the sentence was re-phrased:
New L557-564: “Although the chickpea beverage has the smallest particles, its sedimentation was remarkably faster compared to the other beverages, which may be significantly affected by the lower viscosity of the solution and higher bulk density of its particles. As shown in Table 4, the chickpea beverage presents the lowest zero shear-rate viscosity in comparison to the lupin and their mixture-based beverages. Therefore the dispersion viscosity should not be neglected, nor the fact that the bulk density of chickpea is, on average, bigger than lupin [52].”
A new link-reference was added:
[52] Table of typical grain bulk densities and angles of repose. Available online: http://www.leoncooksilos.com.au/Typical%20Grain%20Bulk%20Densities%20and%20Angles%20of%20Repose.pdf (accessed on 17.09.2019)
11-L323 This graph (Fig. 4) is hard to read for comparison of individual samples.
New L784-785: The lines were enlarged accordingly, for each flow curve.

Reviewer 2 Report
The focus point of this manuscript is interesting and seats in an area with high industry value. However, this manuscript comes up with number of limitations. I have attached the file for some examples. The major comments are listed here: 1. The manuscript writing/presentation style is poor. I am confused with the objective of the manuscript. By reading the manuscript it looks like the objective was sequential optimisation of the pulse beverage/milk processing. However, it has not been clearly mentioned in the abstract or introduction. In those sections it appeared as the comparison of different processing methods. Thus, authors need to clearly mention their objective and present the results according to that.
2. The abstract is poorly written. There is no clear presentation of the findings. No quantitative comparison of the methods. It needs to be improved significantly.
3. Assuming the main focus on developing a novel methodology for pulse beverage, I was expecting a detail discussion/interpretation/limitation of published methodology in the introduction. There is a scope to improve this.
4. In the M&M section, not enough references have been used to justify the methodology. Particularly when the novel approaches have been used.
5. Results and Discussion putting together made this manuscript really hard to follow. The flow of the results is poor, it is particularly a major issue when a sequential beverage processing methodology was optimised.
6. I found authors very rarely interpreted the results of this study with the previously published methodology. It needs to be improved a lot. When a novel processing protocol has been claimed it needs to have a robust discussion/comparison with previous works.
7. There was no proper chemical analysis. Only acidity has been measured. Considering an alternative of the milk further chemical analysis is recommended as for examples, macro/micro nutrients, amino acids etc. There was no indication of the allergenic proteins or compounds.
8. Overall, the manuscript presentation has reduced significantly the value of the research work has been carried out. Authors may consider a simple and clear presentation of the work, with a robust discussion.
Author Response
Carla Margarida Duarte
Instituto Superior de Agronomia
Tapada da Ajuda
1349-017, Lisboa, Portugal
Tel. (+351) 21 365 31 00
Fax: (+351) 21 365 31 95
E-mail: carladuarte@isa.ulisboa.pt
Dear Sirs,
We are pleased to re-submit herewith our revised manuscript entitled “Legume beverages from chickpea and lupin, as new milk alternatives” with Foods reference 933617.
We would like to thank the Referees for their constructive comments and criticism on the manuscript. All their suggestions were considered and included in the attached revised manuscript (main changes highlighted in different colour). The answers to the individual comments/questions are provided as attachment and we believe have significantly increased the quality of the manuscript.
Looking forward to hearing from you in due course.
Yours sincerely,
Carla Margarida Duarte (on behalf of all coauthors)
REVIEWER 2
Comments and Suggestions for Authors
The focus point of this manuscript is interesting and seats in an area with high industry value. However, this manuscript comes up with number of limitations. I have attached the file for some examples. The major comments are listed here:
- The manuscript writing/presentation style is poor. I am confused with the objective of the manuscript. By reading the manuscript it looks like the objective was sequential optimization of the pulse beverage/milk processing. However, it has not been clearly mentioned in the abstract or introduction. In those sections it appeared as the comparison of different processing methods. Thus, authors need to clearly mention their objective and present the results according to that.
The abstract was reviewed as requested (new L18-32): “Recently, milk consumption has been declining and there is a high demand for non-dairy beverages. However, market offers are mainly cereal and nut-based beverages, which are essentially poor in protein (typically, less than 1.5% against the 3.5% in milk) and are not true milk replacers in that sense.
In this work, new beverages from different pulses (pea, chickpea and lupin) were developed using technologies enabling the incorporation of a high level of seed components, with low or no discharge of by-products. Different processing steps were sequentially tested and discussed for the optimization of the sensorial features and stability of the beverage, taking into account the current commercial non-dairy beverages. The lupin beverage protein contents ranged from 1.8 to 2.4% and the chickpea beverage varied between 1.0 and 1.5%. The “milk” yield of the optimized procedure B was 1121 g/100 g of dry seed and 1247 g/100 g of dry seed, for chickpea beverage and lupin beverage, respectively. Sensory results shows that chickpea beverage with cooking water had the better taste. All pulse-based beverages are typical non-Newtonian fluids, similarly to current non-dairy alternative beverages. In this respect, the sprouted chickpea beverage, without the cooking water, presents the most pronounced shear-thinning behavior of all formulations.”
The sentence in the Introduction section was re-phrased (new L152-161): “The main goal of this work was to develop pulse beverages from pea, chickpea and lupin seeds (and their mixtures) focusing on the best technological options to obtain a high-protein beverage (higher than 1.5%) with reduced “beany” flavor, with the less possible discharge of by-products. So, a sequence of different processing steps were tested and discussed for the optimization of the sensorial features and beverage´s stability. Samples were compared considering the relevant chemical parameters in each process step, such as the total and volatile acidities, protein and carbohydrates content, including starch and glucose. Moreover, the particle size, beverage color and sensory evaluation were also accessed accordingly. The rheological flow behavior of the developed pulse-based beverages was compared with eight commercial non-dairy beverages, selected from a previous study [39], in order to serve as a guide for the preferred consumers mouthfeel and texture.”
- The abstract is poorly written. There is no clear presentation of the findings. No quantitative comparison of the methods. It needs to be improved significantly.
The reviewed abstract is exposed in the previous comment/suggestion 1.
- Assuming the main focus on developing a novel methodology for pulse beverage, I was expecting a detail discussion/interpretation/limitation of published methodology in the introduction. There is a scope to improve this.
The Introduction section was extensively revised taking into account your valuable suggestions, and the discussion of published methodologies has also been improved to meet the reviewer concerns.
All reference´s numbers in the manuscript and the list counting were corrected accordingly.
- In the M&M section, not enough references have been used to justify the methodology. Particularly when the novel approaches have been used.
Following the reviewer comments, new suitable references for soaking, pressure cooking, colloidal milling, pasteurization and germination have been added:
New L172-176: “The beverage’s production evolved into the following final optimization (Figure 1(a)): 150 g of dried seeds were soaked twice in warm tap water (30-35ºC) and once in cold tap water (15-20 ºC) for ca. 16 h [32]. All soaking waters were discarded. Then, the soaked seeds were cooked for 30 min after boiling in a pressure pan with 1.5 L of fresh tap water [31].”
New L252-256: “The milling step adapted from previous studies [21,40], included grinding the seeds (or sprouts) with only 200 ml of the water (cooking or new), in a food processor (Bimby-Worwerk, Germany) at 20.500 rpm, for 4 min, followed by colloidal milling performed by a mortar grinder, at 70 rpm, for 15 min. (lab scale) using the remaining volume of water (cooking or new). All beverages were sieved with a strainer before being bottled in sterilized flasks (100 ºC, 10 min).”
New L257-259: “In the pasteurization step (adapted from a previous study [40]), the capsulated filled flasks (beverage temperature > 90 ºC), were submitted to a thermal shock, inside the pressure cooker for 1 min, in boiling water.”
The sentence was re-phrased (new L260-262): “The germination step was adapted from a previous study [37] and included 2 days of incubation where the moist seeds were kept inside an open sterilized flask, at room temperature and protected from light and dust with a cloth.“
New references were added:
[37] Murugkar, D.A. Effect of sprouting of soybean on the chemical composition and quality of soymilk and tofu. J Food Sci Technol 2014, 51, 915–921. https://dx.doi.org/10.1007/s13197-011-0576-9
[40] Nelson, A.I.; Steinberg, M.P.; Wei, L.S. Illinois process for preparation of soymilk. J Food Sci 1976, 41, 57–61. https://doi.org/10.1111/j.1365-2621.1976.tb01100.x
- Results and Discussion putting together made this manuscript really hard to follow. The flow of the results is poor, it is particularly a major issue when a sequential beverage processing methodology was optimized.
The Results and Discussion sections were separated and improved as requested.
In the Results section the following sentences were added or re-phrased to help better understanding the reasons why each process step was introduced or improved. Also the interpretation of the results was supported, when justifiable, with the previously published methodology/ technological knowledge references (reviewer 2: part of next suggestion 6):
New L396-398: “The initial experimental trials where performed with 5% to 10% of dried seeds. The goal was to achieve a final beverage with lower viscosity and a protein content between 1-4%. No “okara” was obtained in the following first tests since beverages were not sieved.”
New L399-403: “At the initial testing, lupin-based beverage always presented phase separation with sediment (data not shown) must probably due to high particle size. This is not related to the influence of beverage pH on protein’s functional properties. The lupin-based beverages had a pH value of 6.0 + 0.2, while chickpea and pea-based beverages presented a pH around 6.7. The protein solubility with pH shows a minimum between 4 and 5, but at values of 6 it evidences a good solubility [12,48].”
New L409-415: “Therefore, this undesired gelation phenomena affected their viscosity [21] and hampered the formation of homogeneous and liquid-like beverages. The formed pudding-like gel was more robust in green pea- and yellow pea-based beverages than in chickpea. This is related to the higher content of starch, around 45% in pea and chickpea, compared to 6.7% in lupin [49]. This is expected to occur due to the heating-induce breakdown of the amylose and amylopectin intermolecular association, resulting in high viscous solutions and, eventually, changing into a strong gel (retrogradation) upon prolonged storage [50]. This gelation drawback made us retreat the use of peas.”
New L418-423: “Chickpea- and lupin-based beverages still evidenced a slight “beany” flavor at this stage of processing progression. Additional adjustments on the processing steps were considered: the cooking step was performed in pressure pan instead of food processor; the sieving step was introduced after milling to remove major particles; and the peeling step (before and after cooking) was also introduced, knowing that the last can contribute to reduce the bitter taste attributed to husk’s phenolic compounds [28,33].”
New L430-432: “However, the processing with cooking water or with new water evidenced significant differences in pH values between chickpea pulse-based beverages (Table 1), maybe due to the slight alkaline tap water (pH 7-8) used.”
A reference was introduced (new L422-423): “Stability towards phase separation can be improved by reducing the particle size and/or the use of rheological modifiers [5].”
New L514-518: “To confirm process suitability (pressure cooking and sieving) to keep the expected protein content, the beverages were analyzed accordingly. The lupin beverages evidenced significant higher values (1.8-2.4%) when compared to the chickpea beverages (1.0-1.5%) (Table 2). Both pulse beverages produced with cooking water presented higher protein values compared to new water, thus confirming the protein solubilization into water during the cooking step.”
New L668-672: “This new way of milling was used before [40] in a developed process for lupin beverage, in which an additional step of colloidal milling was introduced in stablished soy beverage fabrication process, in order to improve the dispersion stability of the lupin particles in the liquid phase [21]. The “okara" was further reduced from 16-25% to 1.8-6.5%, evidencing the efficacy of particles’ dimension reduction.”
New L679-687: “The reason for the observed gelation, may rely on the extended duration and heating during colloidal milling, which allowed the gelling of the starch contained in both chickpea seed and cooking water [16]. Accordingly, the lupin, chickpea and their mixture, produced with the cooking waters, displayed the higher starch contents when compared to beverages produced with new water. As expected, the pulse beverages produced from sprouts did not gel due to starch hydrolysis during the germination of seeds. This observation was supported by lower starch contents in lupin sprouts and chickpea sprouts-based beverages when compared to the corresponded non-germinated counterparts.”
New L757-762: “Although the chickpea beverage has the smallest particles, its sedimentation was remarkably faster compared to the other beverages, which may be significantly affected by the lower viscosity of the solution and higher bulk density of its particles. As shown in Table 4, the chickpea beverage presents the lowest zero shear-rate viscosity in comparison to the lupin and their mixture-based beverages. Therefore the dispersion viscosity should not be neglected, nor the fact that the bulk density of chickpea is, on average, bigger than lupin [52].”
A new sentence was added (new L816-817): “This is a confirmation of the benefits of the germination step in legume beverages with high starch content and cooking water replacement.”
We believe the changes performed met the reviewer concerns and make the manuscript easier to follow.
New references were added:
[12] Piornos, J.A; Burgos-Díaz, C.; Ogura, T.; Morales, E.; Rubilar, M.; Maureira-Butler, I.; Salvo-Garrido, H. Functional and physicochemical properties of a protein isolate from AluProt-CGNA: A novel protein-rich lupin variety (Lupinus luteus). Food Res Int 2015, 76, 719-724. https://doi.org/10.1016/j.foodres.2015.07.013
[16] Fennema, O.R. Chapter 4: Carbohydrates. In: Food chemistry, 3rd ed.; Publisher: Fennema O.R., University of Wisconsin-Madison, Eds.; Marcel Dekker Inc., New York, 1996; pp. 158-221.
[40] Nelson, A.I.; Steinberg, M.P.; Wei, L.S. Illinois process for preparation of soymilk. J Food Sci 1976, 41, 57–61. https://doi.org/10.1111/j.1365-2621.1976.tb01100.x
[48] Sousa, I.M.N.; Morgan, P.J.; Mitchell, J.R.; Harding, S.E.; Hill, S.E. Hydrodynamic characterization of lupin proteins: solubility, intrinsic viscosity, and molar mass. J Agr Food Chem 1996, 44, 3018-3021. http://dx.doi.org/10.1021/jf950516f
[50] Tako, M.; Tamaki, Y.; Teruya, T.; Takeda, Y.. The Principles of Starch Gelatinization and Retrogradation. Food Nutr Sci 2014, 5, 280-291. http://www.scirp.org/journal/fns) http://dx.doi.org/10.4236/fns.2014.53035
[52] Table of typical grain bulk densities and angles of repose. Available online: http://www.leoncooksilos.com.au/Typical%20Grain%20Bulk%20Densities%20and%20Angles%20of%20Repose.pdf (accessed on 17.09.2019)
- I found authors very rarely interpreted the results of this study with the previously published methodology. It needs to be improved a lot.
This suggestion has been taken into account (see the previous reply on #5 comment/ suggestion.
When a novel processing protocol has been claimed it needs to have a robust discussion/comparison with previous works.
We agree with the reviewer and the Discussion section was revised accordingly as requested.
- There was no proper chemical analysis. Only acidity has been measured. Considering an alternative of the milk further chemical analysis is recommended as for examples, macro/micro nutrients, amino acids etc. There was no indication of the allergenic proteins or compounds.
Although we agree with the reviewer comment, the intent of the analysis performed during the fabrication process progression was to evaluate the suitability of each process step to lead at pulse beverages with the highest content of protein, without “beany flavor”, good sensorial features and stability. Indeed, the more complex evaluation of micro nutrients and bioactive components are foreseen in further future beverage’s analysis.
Nevertheless, a new chemical analysis was included (new L301-303): “The carbohydrates content of the optimized pulse beverages was carried out according to Dubois et al. [45]. The analysis were performed in triplicate and data expressed as g of carbohydrates per 100 mL of pulse beverage.”
A new method description was introduced (L304-312): “2.2.4 High performance liquid chromatography (HPLC) analysis. The D-glucose content of the optimized pulse beverages (Procedure B) was obtained by HPLC [46]. Briefly, 2 mL of each sample was centrifuged at 12.000 rpm for 10 min, and 100 µL of supernatant was collected. After its dilution in H2SO4 (98%, Merck) (50 mM) (1:10 (v/v)), the samples were centrifuged (12.000 rpm, 10 min) to discard the precipitated protein, and filtered through a 0.20 μm-pore-size filter (Whatman). D-glucose was quantified in a high-performance liquid chromatography system (Waters) equipped with a refractive index detector (Waters 2414) and a RezexTM ROA Organic Acid H+ (8%) column (300 mm × 7.8 mm, Phenomenex), at 65 °C. H2SO4 (5 mM) was used as mobile phase at 0.5 mL.min-1. Results are expressed as g of glucose per 100 mL of pulse beverage.”
The sentence was re-phrased (L694-695): “The particle size distribution of the pulse-based beverages from procedure B with new water (B2) was analyzed and also their carbohydrates, starch and glucose contents (Table 3).”
A new table was introduced (new L763-766): “Table 3. Carbohydrates, including starch and glucose contents in pulse beverages obtained from the processing procedure B - whole seeds, colloidal milling and without the cooking waters.”
A new paragraph was added (new L767-779): “The carbohydrates, starch and glucose contents of the optimized beverages from Procedure B were also analyzed to infer on how much carbohydrates remained after discarding the cooking waters (replaced by new one). The highest values for carbohydrates and starch were evidenced in the chickpea beverage and the lowest in lupin´s (Table 3), accordingly to their composition. Comparing these results to the correspondent nutritional composition of dryed seeds [49], the reference values for carbohydrates in beverages with 10% of dryed seeds (10 g/100 mL) are: 5.6 g/100 mL for chickpea and 1 g/100 mL for lupin are lower than those presented in Table 3. For sugars (glucose included) the reference values [49] are 0.3 g/100 mL for chickpea and 0.05 g/100 mL for lupin, comparing to the calculated values (Table 3) 0.45 and 0.06 respectively, which are similar and differences can be attributed to different plant varieties and edapho-climatic conditions. The starch values expected for 10 grams of dried seeds per 100 mL, are 4.5 g for chickpea and 0.7 g for lupin [49], the lower values obtained, 0.689 g and 0.006 g, respectively, confirm the elution of starch from beverages during processing/ cooking.”
New references were added:
[45] Dubois, M.; Gilles, K.; Hamilton, J.K.; Rebers, P.A.; Smith, F. A colorimetric method for the determination of sugars and related substances. Nature 1951, 168, 167. https://doi.org/10.1038/168167a0
[46] Santos, M.V.; Faria, N.T.; Fonseca, C.; Ferreira, F.C. Production of mannosylerythritol lipids from lignocellulose hydrolysates: tolerance thresholds of Moesziomyces antarcticus to inhibitors. J Chem Technol Biotechnol 2019, 94, 1064–107. https://doi.org/10.1002/jctb.5853
- Overall, the manuscript presentation has reduced significantly the value of the research work has been carried out. Authors may consider a simple and clear presentation of the work, with a robust discussion.
Following the reviewer comment, the manuscript and all sections previously indicated were extensively revised. We believe the changes performed met the reviewer concerns and make the manuscript clear and easier to follow.
The Funding and Acknowledgments sections were also revised to translate the correct funding of this experimental work:
New L893: “Funding: This work was supported by the FCT Project PTDC/BAA-AGR/28370/2017: “Legumilk”.”
New L894-897: “Acknowledgments: Authors acknowledge FCT/Portugal for financing the projects PTDC/BAA-AGR/28370/2017 and PTDC/ASP-SIL/30619/2017, the research unit UID/AGR/04129/LEAF, the researcher grant CEECIND/01014/2018 and the donation of pulse seeds by the Portuguese research institute INIAV (Elvas, Portugal).”

Round 2
Reviewer 1 Report
The authors have improved their manuscript considerately and made some additional efforts in better characterizing the pulses and pulse-beverages (e.g. determination of carbohydrate content). Also, the aim better matches the results as described in this manuscript. However, I believe that this manuscript is still needs attention on the following points.
- The underlying mechanisms are still poorly explained or hypothesized on in the Discussion. For instance, why do the beverages display shear-thinning behaviour? Is there an influence of pH or composition? Do the authors hypothesize aggregate break-up or rather alignment with flow? Is the mechanism similar to the mechanism causing shear-thinning in cow milk?
- Also, the manuscript would benefit from being more specific sometimes. In L283 the authors refer to literature and state that lupin is well soluble at pH 6, because the isoelectric point is somewhere between 4-5, but more accurate values could be given.
- Finally, a clear conclusion i.e. stating the main outcome of this research would be to the benefit of the reader.
Small comments regarding spelling:
- L89: technic should read technique
- L131: less should read least
Author Response
Carla Margarida Duarte
Instituto Superior de Agronomia
Tapada da Ajuda
1349-017, Lisboa, Portugal
Tel. (+351) 21 365 31 00
Fax: (+351) 21 365 31 95
E-mail: carladuarte@isa.ulisboa.pt
Dear Sirs,
We are pleased to re-submit herewith our revised manuscript entitled “Legume beverages from chickpea and lupin, as new milk alternatives” with Foods reference 933617.
We would like to thank the Referee for her/his constructive comments and criticism on the manuscript. All suggestions were considered and the answers to the individual comments/questions are provided as attachment to this cover letter.
Looking forward to hearing from you in due course.
Yours sincerely,
Carla Margarida Duarte (on behalf of all coauthors)
REVIEWER 1
Comments and Suggestions for Authors
The authors have improved their manuscript considerately and made some additional efforts in better characterizing the pulses and pulse-beverages (e.g. determination of carbohydrate content). Also, the aim better matches the results as described in this manuscript. However, I believe that this manuscript is still needs attention on the following points.
- The underlying mechanisms are still poorly explained or hypothesized on in the Discussion. For instance, why do the beverages display shear-thinning behaviour? Is there an influence of pH or composition? Do the authors hypothesize aggregate break-up or rather alignment with flow? Is the mechanism similar to the mechanism causing shear-thinning in cow milk?
This is an important point raised by the reviewer. A new sentence was added in the Discussion section (L746-749): “This behavior is common in these kind of fluids with complex composition and particles. The explanation is either the alignment of asymmetrical particles to flow and rate of breakage of interactions between molecules, higher than the formation of molecules low energy interactions, like physical entanglements.”
- Also, the manuscript would benefit from being more specific sometimes. In L283 the authors refer to literature and state that lupin is well soluble at pH 6, because the isoelectric point is somewhere between 4-5, but more accurate values could be given.
We used the adequate references that support this pH range and protein solubility at pH 6. Nevertheless, the sentences were re-phrased:
New L84-86: “Proteins from different pulses generally show higher solubility at alkaline and acidic pH values (pH < 4.0 and pH > 5.0), whereas they are less soluble at pH around their isoelectric point (4.6 for lupin proteins and 4.5 for pea and chickpea proteins) [12,13].”
New L376-379: “The protein solubility with pH shows a minimum at pI values (4.6 and 4.5, respectively), but at values of 6 it evidences a good solubility [12,13,48].”
- Finally, a clear conclusion i.e. stating the main outcome of this research would be to the benefit of the reader.
In Discussion section the following sentences already state the main outcome of this research:
L750-754: “Overall, the strategies adopted and the optimization steps performed in this work led to the development of novel pulse-based beverages with several appealing features (e.g., protein content, rheological behavior, color and appearance) that are highly competitive for the current commercial non-dairy beverages. Therefore, this work contributes to pave the way for the development of novel pulse-based beverages as viable alternatives for cow milk.”
Small comments regarding spelling:
- L89: technic should read technique
- L131: less should read least
The manuscript was revised for spelling errors as requested.

Reviewer 2 Report
Authors have improved the manuscript significantly. It is now better to understand many parts, for example ABSTRACT and INTRODUCTION. However, there are still some concerns which need to fixed.
- The writing style of english is poor even in many places in is not correct. I have just picked a few examples in the attached file. I strongly recommend that the manuscript should be checked carefully for writing.
- Authors have separated the RESULT and DISCUSSION sections, which is good. However, the Discussion part is really substandard. It needs to improve a lot. In contrast, there are many statements, references in the result sections which are not usual either. Authors should shift the discussions of the results from the RESULT section to the DISCUSSION section. I have picked couple of examples in the attached file. But authors should check carefully the whole section for the similar things.
- The conclusion of the manuscript is still not very rigid or specific. Authors need to improve this section by providing the novel outputs of this study and how this can be utilised in the industry or in further research.

Author Response
Carla Margarida Duarte
Instituto Superior de Agronomia
Tapada da Ajuda
1349-017, Lisboa, Portugal
Tel. (+351) 21 365 31 00
Fax: (+351) 21 365 31 95
E-mail: carladuarte@isa.ulisboa.pt
Dear Sirs,
We are pleased to re-submit herewith our revised manuscript entitled “Legume beverages from chickpea and lupin, as new milk alternatives” with Foods reference 933617.
We would like to thank the Referee for her/his constructive comments and criticism on the manuscript. All suggestions were considered and the answers to the individual comments/questions are provided as attachment to this cover letter.
Looking forward to hearing from you in due course.
Yours sincerely,
Carla Margarida Duarte (on behalf of all coauthors)
REVIEWER 2
Comments and Suggestions for Authors
Authors have improved the manuscript significantly. It is now better to understand many parts, for example ABSTRACT and INTRODUCTION. However, there are still some concerns which need to fixed.
- The writing style of english is poor even in many places in is not correct. I have just picked a few examples in the attached file. I strongly recommend that the manuscript should be checked carefully for writing.
The entire manuscript was carefully revised for writing style and the spelling errors were checked with the use of Ginger software, as requested.
The sentence was re-phrased (L250-251):” The initial experimental trials were performed with 5% to 10% (w/v) of dried seeds in the beverage, to achieve a lower viscosity and a protein content between 1-4% (w/v).”
The sentence was re-phrased (L253-254): “At the initial testing, lupin-based beverage always presented phase separation with sediment (data not shown) probably due to high particle size.”
The sentence was re-phrased (L255-256): “Lupin-based beverages had a pH value of 6.0 + 0.2, while chickpea and pea-based beverages presented a pH around 6.7.”
The sentences were re-phrased (L363-367): “The particle sedimentation was analyzed during 24 h and their macroscopic evolution was recorded over time. The onset of particle sedimentation of the chickpea beverage was visible after 15 min of incubation and complete sedimentation occurred after ca. 24 h. Instead, for lupin and lupin + chickpea beverages, the sedimentation started after 24 h and occurred, gradually, during ca. 6 days. The optimization of the milling step enhanced the stability of beverages.”
- Authors have separated the RESULT and DISCUSSION sections, which is good. However, the Discussion part is really substandard. It needs to improve a lot. In contrast, there are many statements, references in the result sections which are not usual either. Authors should shift the discussions of the results from the RESULT section to the DISCUSSION section. I have picked couple of examples in the attached file. But authors should check carefully the whole section for the similar things.
We appreciate the reviewer suggestion, but from MDPI Foods paper structure, the RESULTS section “should provide a concise and precise description of the experimental results, their interpretation as well as the experimental conclusions that can be drawn”. The results interpretation and experimental conclusions (not discussion/ comparison with other works) were exposed by the authors in RESULTS section, and some references were included to support them, as requested previously by Reviewer 1 (Review Reports round 1 and 2) and Reviewer 2 in the Review Report round 1.
Nevertheless, three sentences from the RESULTS section were eliminated (explained below), and one sentence that discuss the results and how they can be interpreted in perspective of previous studies, was moved to the DISCUSSION section:
New L446-448: “This observation is supported by the recent work of Niño and coworkers [49], which have determined a protein content in the husks of chickpea of ca. 4.5%. This value corresponds proportionally to the lower content (0.72 to 0.86%) estimated in this work for the entire chickpea seed.”
The reference [51] was re-numbered to [49]: Niño-Medina, G.; Muy-Rangel, D.; Garza-Juárez, A.J.; Vázquez-Rodríguez, J.A.; Méndez-Zamora, G.; Urías-Orona, V. Composición nutricional, compuestos fenólicos y capacidad antioxidante de cascarilla de garbanzo (Cicer arietinum). Archivos Latinoamericanos Nutrición 2017, 67(1), 68-73. Available online: https://www.alanrevista.org/ediciones/2017/1/art-10/
Adding to that, the DISCUSSION section has already exposed some comparisons of our main results with previous works:
L451-459: “The “milk” yields obtained for chickpea and lupin beverages during the optimized procedure B, focused on lower discharge of by-products, were around 12.2-12.5 kg of beverage per kg of seed, respectively, which is ca. 4 times higher in comparison to the “milk” yield obtained for oat beverage (2.85 kg/ kg of rolled oat) developed by Deswal and co-workers [50] with a technological process using enzymatic hydrolysis to liquefy oat’s higher content of starch. On the other hand, the slightly alkaline tap water (pH 7-8) used for soaking and cooking steps may have also contributed to this higher “milk” yield values. A previous study [51] has shown that the alkali soaking has significantly improved the yield of total solids in sesame beverage, which is due to the higher solubility of the plant proteins at these pH values.”
The following sentences in the RESULTS section were eliminated:
- This suggestion for stability improvement was already mentioned in the Discussion section (old L332-333): “Stability towards phase separation could be improved by reducing the particle size and/or the use of rheological modifiers [5].”
- The two references for colloidal milling ([21] and [40]) were already mentioned in M&M section (old L372-375): “This new way of milling was used before [40] in a developed process for lupin beverage, in which an additional step of colloidal milling was introduced in stablished soy beverage fabrication process, in order to improve the dispersion stability of the lupin particles in the liquid phase [21].”
- This suggestion was already mentioned in the Discussion section (old L458-459): “This suggests that, from a rheological point of view, the developed beverages in this work are very promising.”
The last paragraph of the DISCUSSION section was moved to the CONCLUSIONS section:
- “Overall, the strategies adopted and the optimization steps performed in this work led to the development of novel pulse-based beverages with several appealing features (e.g., protein content, rheological behavior, color and appearance) that are highly competitive for the current commercial non-dairy beverages. Therefore, this work contributes to pave the way for the development of novel pulse-based beverages as viable alternatives for cow milk. Nevertheless, further optimizations must be performed in future works, such as considering the use of enzymes, to refine the beverage mouthfeel, the addition of natural flavors, for an improved and pleasant sensorial perception, and the use of high pressure homogenization to reduce particle´s dimensions, improving beverages stabilization and microbial shelf life extension.”
- The conclusion of the manuscript is still not very rigid or specific. Authors need to improve this section by providing the novel outputs of this study and how this can be utilized in the industry or in further research.
The CONCLUSIONS section was introduced into the manuscript:
L471-487: “5. Conclusions. Legume beverages present the most balanced composition and a protein content similar to cow milk, but face technological issues often related to processing or preservation. Heat treatment, such as cooking and pasteurization, are able to remove off-flavors, the most challenging barrier to consumer acceptance, however, high temperatures may cause excessive protein denaturation, lower protein solubility and may increase legume beverage viscosity, affecting its stability. The colloidal milling, on the other hand, is one technological intervention capable to increase the beverage physical stability by reducing the size of dispersed phase particles. In this work, several of these issues have been tackled and, overall, the processing strategies adopted and the optimizations performed led to the development of novel pulse-based beverages with several appealing features (e.g., protein content, rheological behavior, color and appearance) that are highly competitive in the current commercial non-dairy beverages. Therefore, we believe this work strongly contributes to pave the way for the development of novel pulse-based beverages as viable alternatives for cow milk. Nevertheless, further future optimizations must be performed, such as considering the use of enzymes, to refine the beverage mouthfeel, the addition of natural flavors, for an improved and pleasant sensorial perception, and the use of high pressure homogenization to reduce particle´s dimensions, improving beverages stabilization and microbial shelf life extension.”
